# Development of INER-PP-F11N as the Peptide-Radionuclide Conjugate Drug Against CCK2 Receptor-Overexpressing Tumors

**DOI:** 10.3390/ijms26146565

**Published:** 2025-07-08

**Authors:** Ming-Cheng Chang, Chun-Tang Chen, Ping-Fang Chiang, I-Chung Tang, Cheng-Liang Peng, Yuh-Feng Wang, Yi-Jou Tai, Ying-Cheng Chiang

**Affiliations:** 1Department of Isotope Application Research, National Atomic Research Institute, P.O. Box 3-27, Longtan, Taoyuan 325207, Taiwan; mcchang@nari.org.tw (M.-C.C.); ctchen@nari.org.tw (C.-T.C.); ckdopamine@nari.gov.tw (P.-F.C.); ictang@nari.org.tw (I.-C.T.); clpeng@nari.org.tw (C.-L.P.); 2Department of Nuclear Medicine, Taipei Veterans General Hospital, Taipei 112201, Taiwan; yfwang6@vghtpe.gov.tw; 3Department of Biomedical Imaging and Radiological Sciences, National Yang Ming Chiao Tung University, Taipei 112304, Taiwan; 4Department of Medical Imaging and Radiological Technology, Yuanpei University of Medical Technology, Hsin-Chu City 300102, Taiwan; 5Department of Obstetrics and Gynecology, College of Medicine, National Taiwan University, Taipei 100226, Taiwan; 6Department of Obstetrics and Gynecology, National Taiwan University Hospital, Taipei 100226, Taiwan; 7Department of Obstetrics and Gynecology, National Taiwan University Hospital, Hsin-Chu Branch, Hsin-Chu City 300195, Taiwan

**Keywords:** INER-PP-F11N, CCK2R, theragnostic, radiopharmaceutical agent, peptide-radionuclide conjugate drug

## Abstract

This work aimed to evaluate two albumin affinity structure-containing peptide-radionuclide conjugate drugs, INER-PP-F11N-1 and INER-PP-F11N-2, for the diagnosis/treatment of cholecystokinin receptor subtype 2 (CCK2R)-overexpressing cancers. We developed In-111- and Lu-177-labeled INER-PP-F11N radiopharmaceuticals and compared them with the current PP-F11N to investigate metabolic stability, biodistribution, SPECT/CT imaging, and therapeutic responses in CCK2R-expressing tumor xenograft mice. The metabolic stability of [^111^In]In/[^177^Lu]Lu-INER-PP-F11N remained above 90% for up to 144 h after labeling, indicating that the compound is highly stable under in vitro conditions. INER-PP-F11N showed 27% and 11% higher cellular uptake and internalization than PP-F11N, respectively. In vivo SPECT/CT imaging confirmed that INER-PP-F11N could accumulate at the tumor site of mice 24 h after receiving the two radiopharmaceutical agents. Biodistribution analysis revealed a significantly greater tumor uptake and reduced accumulation of INER-PP-F11N in the kidneys compared with PP-F11N. Furthermore, INER-PP-F11N significantly inhibited the growth of the CCK2R-overexpressing tumors in mice. The INER-PP-F11N radiopharmaceutical was superior as a theragnostic agent compared with the current PP-F11N. Our study suggests that INER-PP-F11N may be an innovative radiopharmaceutical agent for CCK2R-overexpressing tumors.

## 1. Introduction

The cholecystokinin-2 receptor (CCK2R) is a peptide-binding G-protein-coupled receptor (GPCR) widely distributed in the gastrointestinal tract and central nervous system. It exerts its biological effects through activation of the phosphatidylinositol–calcium second messenger pathway in response to gastrin and cholecystokinin [1,2]. CCK2R has been identified in gastrointestinal smooth muscle cells, pancreatic cells, myenteric neurons, and immune cells, where it plays a regulatory role in cellular maturation, proliferation, and migration [3,4,5,6]. Due to its involvement in key physiological processes and its limited expression in most normal tissues, CCK2R has emerged as a promising molecular target for cancer imaging and therapy. Notably, CCK2R is overexpressed in a variety of malignancies, including gastric cancer, medullary thyroid cancer, colorectal cancer, small cell lung cancer, and pancreatic cancer [7,8,9,10,11,12,13,14,15,16]. Over 90% of medullary thyroid cancers express CCK2R, highlighting its potential for targeted diagnostics and radionuclide therapy [17,18,19].

Several CCK2R-targeting ligands, including agonists and antagonists, have been developed. Exogenous gastrin-induced proliferation in human pancreatic cancer cells can be inhibited by the CCK2R antagonist, L-365260 [20,21]. Another cholecystokinin antagonist, CI-988 (PD-134308), has been shown to block gastrin binding to CCK2R and improve survival in mouse models inoculated with human gastric cancer ascites cells [22,23]. Moreover, other small-molecule inhibitors such as dibutyryl cGMP [1], CR-1409 (lorglumide) [24], and devazepide [25] have also been explored. Despite these efforts, the clinical application of CCK2R-targeted drugs remains limited, and refinement of pharmacologic agents is still needed. In the field of nuclear medicine, both past and ongoing efforts have focused on radiolabeled peptide analogs of CCK2R. PP-F11 and its derivative PP-F11N have shown promise in clinical investigations [26]. However, significant limitations include high renal retention, low in vivo stability, and toxicity. For instance, treatment with Y-90-labeled CCK2R analogs resulted in severe renal toxicity in 25% of patients, restricting their therapeutic use [27]. These challenges underscore the need for improved radiopharmaceutical designs with enhanced targeting efficiency and reduced off-target toxicity.

To address these limitations, we developed two novel DOTA-conjugated CCK2R-targeting peptide analogs, DOTA-INER-PP-F11N-1 and DOTA-INER-PP-F11N-2. These compounds aim to improve tumor specificity and minimize renal accumulation, thereby enhancing the efficacy and safety of CCK2R-targeted radionuclide therapy.

## 2. Results

### 2.1. Structural Designs of Various CCK2R Derivatives Between DOTA and the C-Terminal Binding Sequence

Previous clinical investigations demonstrated that Lu-177-PP-F11N could be a promising radiopharmaceutical agent against medullary thyroid carcinoma. However, high renal and gastrointestinal toxicity and low tumor accumulation limited its clinical application. We modified the existing drug structure by adding the albumin-binding motifs Lys(-4-TBA)-6-AMBA(INER-PP-F11N-1) or Lys(-4-TBA)-6-AHA(INER-PP-F11N-2) to reduce accumulation in non-target organs and enhance tumor-targeting efficacy (Figure 1). DOTA was used to conjugate different ligands and chelate radionuclides such as Lu-177 and evaluate in vitro and in vivo radiopharmaceutical properties.

### 2.2. Radiolabeling DOTA-CCK2R Derivatives Revealed High Efficiency and Reliable Stability

DOTA-PP-F11N, DOTA-INER-PP-F11N-1, and DOTA-INER-PP-F11N-2 were labeled with In-111 and Lu-177 for further experiments. The detection of free In-111 after ITLC extension was defined as 0% radiolabeling efficiency. All the compounds—DOTA-PP-F11N, DOTA-INER-PP-F11N-1, and DOTA-INER-PP-F11N-2—labeled with In-111 showed high labeling efficiency, with initial labeling efficiencies of 98.7 ± 1.6%, 98.2 ± 1.2%, and 98.6 ± 1.0%, respectively (Figure 2A). The metabolic stability of these radiotracers was further evaluated under various conditions, such as PBS at 4 °C or FBS at 37 °C for 0, 4, 24, and 48 h to mimic drug storage/delivery conditions or hosts receiving radiopharmaceutical injections. All [^111^In]In-DOTA-PP-F11N, [^111^In]In-DOTA-INER-PP-F11N-1, and [^111^In]In-DOTA-INER-PP-F11N-2 showed similar metabolic stability at 4 °C (Figure 2A).

Both [^111^In]In-DOTA-INER-PP-F11N-1 and [^111^In]In-DOTA-INER-PP-F11N-2 had more stable properties than the original structure. In contrast to the stable state at 4 °C PBS, DOTA-PP-F11N conjugated with In-111 demonstrated less metabolic stability at 37 °C FBS (Figure 2B). [^111^In]In-DOTA-PP-F11N, [^111^In]In-DOTA-INER-PP-F11N-1, and [^111^In]In-DOTA-INER-PP-F11N-2 showed reliable radiopharmaceutical stability at either 4 °C PBS or 37 °C physiological conditions for 48 h.

DOTA-PP-F11N, DOTA-INER-PP-F11N-1, and DOTA-INER-PP-F11N-2 labeled with Lu-177 were also measured either at 4 °C PBS or 37 °C FBS at each of the time points described previously. DOTA-PP-F11N, DOTA-INER-PP-F11N-1, and DOTA-INER-PP-F11N-2 labeled with Lu-177 showed more than 90% metabolic stability at each time point (Figure 2C). [^177^Lu]Lu-DOTA-PP-F11N, [^177^Lu]Lu-DOTA-INER-PP-F11N-1, and [^177^Lu]Lu-DOTA-INER-PP-F11N-2 showed similar metabolic stability at 4 °C—more than 95% at 4 °C PBS, even 144 h after Lu-177 labeling.

However, DOTA-PP-F11N conjugated with Lu-177 demonstrated instability at 37 °C FBS compared with the stable state at 4 °C PBS. Metabolic stability decreased from 94.6 ± 0.1% at 4 h to 56.4 ± 0.1% at 144 h during the incubation period (Figure 2D). In contrast to [^177^Lu]Lu-DOTA-PP-F11N, two CCK2R derivatives with an albumin-binding domain, including DOTA-INER-PP-F11N-1 and DOTA-INER-PP-F11N-1, showed high metabolic stability even 144 h after Lu-177 labeling in 37 °C FBS. The metabolic stability of both radiopharmaceuticals remained at 99% even 144 h after Lu-177 labeling. These results demonstrate that Lu-177-labeled albumin-binding domains, including AHA or AMBA-containing CCK2R derivatives, were more stable than the original DOTA-PP-F11N under physiological conditions.

### 2.3. Lu-177-Labeled DOTA-CCK2R Derivatives Revealed High Cellular Binding and Internalization in A431-CCK2R (+) but Not A431-CCK2R (−) Cells

The surface binding activities of A431-CCK2R (−) cells treated with [^177^Lu]Lu-DOTA-PP-F11N, [^177^Lu]Lu-DOTA-INER-PP-F11N-1, and [^177^Lu]Lu-DOTA-INER-PP-F11N-2 for 1 h were 3.42 ± 0.21, 1.08 ± 0.49, and 0.59 ± 0.01 (%IA/106 cells), respectively (Figure 3A). Surface binding activity did not significantly increase even when incubation was prolonged to 4 h. The internalization activities of A431-CCK2R (−) cells treated with [^177^Lu]Lu-DOTA-PP-F11N, [^177^Lu]Lu-DOTA-INER-PP-F11N-1, and [^177^Lu]Lu-DOTA-INER-PP-F11N-2 for 1 h were 1.26 ± 0.06, 0.32 ± 0.10, and 0.25 ± 0.04 (%IA/106 cells), respectively. Extending the incubation time to 4 h did not enhance the internalization activity.

We further investigated whether A431-CCK2R (+) cells could take up and internalize these radiopharmaceuticals (Figure 3B). The surface binding activities of A431-CCK2R (+) cells treated with [^177^Lu]Lu-DOTA-PP-F11N, [^177^Lu]Lu-DOTA-INER-PP-F11N-1, and [^177^Lu]Lu-DOTA-INER-PP-F11N-2 increased when the incubation time was increased from 1 to 4 h. Extending the incubation period from 1 to 4 h also enhanced internalization activity. Compared with conventional [^177^Lu]Lu-DOTA-PP-F11N, [^177^Lu]Lu-DOTA-INER-PP-F11N-1 and [^177^Lu]Lu-DOTA-INER-PP-F11N-2 enhanced the internalization activity of A431 CCK2R (+) cells by 26.7% or 13.7%. The results indicated that all Lu-177-labeled CCK2R derivatives could specifically bind to and internalize into CCK2R-expressing cells. Furthermore, [^177^Lu]Lu-DOTA-INER-PP-F11N-1 and [^177^Lu]Lu-DOTA-INER-PP-F11N-2 had more effective binding and internalizing activity in CCK2R-expressing cells compared with conventional [^177^Lu]Lu-DOTA-PP-F11N.

### 2.4. NanoSPECT/CT Imaging of Tumor-Bearing Mice and Biodistribution of Radiotracers

Figure 4A demonstrates the NanoSPECT/CT images of the A431-CCK2R (−)/A431-CCK2R (+) tumor-bearing mice injected with [^177^Lu]Lu-DOTA, [^177^Lu]Lu-DOTA-PP-F11N, [^177^Lu]Lu-DOTA-INER-PP-F11N-1, or [^177^Lu]Lu-DOTA-INER-PP-F11N-2 at various time points. At one hour post-injection, mice administered [^177^Lu]Lu-DOTA exhibited radiotracer accumulation in the urinary bladder, which was rapidly cleared within 24 h after injection. However, mice receiving [^177^Lu]Lu-DOTA-PP-F11N injection displayed radiotracer accumulation in both kidneys from one hour and a delayed washout of more than 48 h after injection. The accumulation of [^177^Lu]Lu-DOTA-PP-F11N in the tumor region was relatively low. In contrast, the radioactivity within the kidneys after [^177^Lu]Lu-DOTA-INER-PP-F11N-1 and [^177^Lu]Lu-DOTA-INER-PP-F11N-2 administration was significantly lower than that after Lu-177-DOPA-PP-F11N administration over the same period. However, imaging of the tumor was limited in mice receiving [^177^Lu]Lu-DOTA-INER-PP-F11N-2 compared with the avid radiotracer uptake within the tumor in mice receiving [^177^Lu]Lu-DOTA-INER-PP-F11N-1. Renal retention of radioactivity was noted in these two radiotracers. These results indicate that these two novel radiopharmaceutical agents had specific targeting of CCK2R-expressing tumors and were eliminated by the urinary system.

The biodistribution of radiolabeled DOTA-CCK2R derivatives using the therapeutic radionuclide Lu-177 is shown in Figure 4B. Radioactivity accumulation was exclusively observed in the tumor and kidneys 72 h after tail-vein administration. Radiotracer accumulation in other organs, such as the heart, spleen, and lungs, was minimal. The intense activity in the kidneys indicated clearance by renal excretion of radiolabeled DOTA-CCK2R derivatives. Tumor uptake of radiolabeled DOTA-CCK2R derivatives remained relatively high, especially for [^177^Lu]Lu-DOTA-INER-PP-F11N-1.

### 2.5. Mice Receiving Lu-177-Labeled DOTA-CCK2R Derivative Therapy Generated Potent Anti-Tumor Effects

Tumor sizes in untreated mice and those treated with [^177^Lu]Lu-DOTA alone (*p* = 0.18, Figure 4C) were similar. Conventional DOTA-CCK2R derivatives [^177^Lu]Lu-DOTA-PP-F11N showed an effective therapeutic response in terms of tumor volume in mice compared with [^177^Lu]Lu-DOTA (tumor volume at D23, 1884.72 ± 333.13 mm^3^ vs. 3237.10 ± 846.25 mm^3^, *p* = 0.048). Mice receiving either [^177^Lu]Lu-DOTA-INER-PP-F11N-1 (841.33 ± 165.70 mm^3^, *p* = 0.023) or [^177^Lu]Lu-DOTA-INER-PP-F11N-2 (1162.60 ± 275.92 mm^3^, *p* = 0.041) also exhibited significantly greater tumor shrinkage compared with those receiving [^177^Lu]Lu-DOTA. Furthermore, tumor volumes in the mice receiving [^177^Lu]Lu-DOTA-PP-F11N, [^177^Lu]Lu-DOTA-INER-PP-F11N-1, and [^177^Lu]Lu-DOTA-INER-PP-F11N-2 treatments were significantly different (*p* = 0.008, one-way ANOVA). Tumor-bearing mice injected with [^177^Lu]Lu-DOTA-INER-PP-F11N-1 had significantly smaller tumor sizes, demonstrating the therapeutic potential of [^177^Lu]Lu-DOTA-INER-PP-F11N.

### 2.6. Histopathological Examination of the Mice After Radiolabeled Drug Administration

Mice treated with [^177^Lu]Lu-DOTA-PP-F11N derivatives were examined using light microscopy (Figure 5). The kidneys from the mice treated with [^177^Lu]Lu-DOTA-PP-F11N showed renal tubular necrosis (arrow 1), moderate renal tubular casts (arrow 2), slight renal tubular degeneration (arrow 3), and neutrophil infiltration (arrow 4). There were no significant histopathological changes in the liver and heart. There were no significant histopathological changes in the kidneys, liver, and heart of the mice treated with [^177^Lu]Lu-DOTA-INER-PP-F11N-1 and [^177^Lu]Lu-DOTA-INER-PP-F11N-2.

## 3. Discussion

This article presents the development and evaluation of two novel albumin-binding CCK2R-targeted radiopharmaceuticals, DOTA-INER-PP-F11N-1 and DOTA-INER-PP-F11N-2, designed to address the key limitations of existing CCK2R-targeted agents such as ^177^Lu-DOTA-PP-F11N, particularly in terms of in vivo stability, tumor selectivity, and renal toxicity. The improved stability observed in [^177^Lu]Lu-DOTA-INER-PP-F11N-1 and -2 compared with the original [^177^Lu]Lu-DOTA-PP-F11N is a key finding of our study. The enhanced radiochemical stability is primarily attributed to the nature of the linker introduced between the DOTA chelator and the peptide moiety: in INER-PP-F11N-1, we employed AMA (aminomethyl-anthranilic acid), while in INER-PP-F11N-2, we utilized AMBA (aminomethylbenzoic acid) as the linker. Both AMA and AMBA are aromatic linkers that introduce increased rigidity and steric hindrance compared with the more flexible and aliphatic linkers used in the original PP-F11N [28]. These structural features help to stabilize the DOTA-Lu(III) complex by reducing local conformational strain or unwanted interactions with adjacent amino acid residues; these structures also provide shielding against transchelation or radiolytic degradation, especially under physiological or oxidative stress conditions [28]. They also alter hydrophilicity and spatial orientation, leading to improved metabolic stability in serum and reduced degradation in biological matrices. This observation is consistent with prior reports indicating that aromatic and sterically protected linkers can significantly improve the in vivo stability of radiolabeled peptides.

Radiolabeling of all three agents (PP-F11N and the two INER derivatives) with In-111 and Lu-177 achieved high initial efficiency (>98%). However, the INER derivatives displayed superior metabolic stability in both PBS and FBS environments. At 144 h post-labeling, the metabolic stability of [^177^Lu]Lu-DOTA-INER-PP-F11N-1 and -2 remained >99% in FBS at 37 °C, whereas that of [^177^Lu]Lu-DOTA-PP-F11N declined below 60%. These findings underscore the stabilizing effect of the AHA and AMBA albumin-binding motifs, in agreement with previous studies showing that albumin affinity prolongs circulation time and reduces degradation [29,30]. The differences in labeling conditions between In-111 and Lu-177 were intentionally optimized based on their distinct coordination chemistries and kinetic profiles. First, the additional shaking step at 500 rpm was applied specifically to the In-111-labeling process to ensure rapid and uniform mixing under mild conditions. ^111^In^3+^ forms relatively fast and stable complexes with DTPA- and DOTA-based chelators at lower temperatures, and vigorous shaking for 15 min at 500 rpm was sufficient to achieve high radiochemical yields (>95%) at 37 °C. In our preliminary experiments, omitting the shaking step resulted in slower complexation and reduced labeling efficiency.

Although albumin binders are generally designed to delay blood clearance, thereby increasing tumor accumulation and reducing renal uptake, some recent reports have emphasized that the actual pharmacokinetic profile depends heavily on binder affinity, molecular linkage, and isotope half-life. For instance, Lu-177-HTK03121 and HTK03123, conjugated with specific albumin-binding motifs, demonstrated prolonged blood retention and very high tumor uptake (up to ~100 %ID/g), in spite of modest renal accumulation [31]. In CA-IX-targeted In-111 tracers bearing albumin binders, early-time biodistribution data show increased blood and tumor activity and reduced kidney uptake compared with non-binder versions [32]. Similarly, TEFAPI-07, a FAPI-04 derivative with an Evans Blue-based albumin-binding moiety, significantly enhanced tumor uptake and retention in pancreatic tumor PDX models [33]. These improvements were strongly correlated with in vitro albumin-binding affinity, suggesting that binding strength modulates blood clearance kinetics. In our study, the unexpectedly rapid blood clearance observed with albumin-binder-containing ligands likely reflects the use of moderate-affinity binders or steric arrangements that attenuate albumin binding. This is consistent with findings that weaker albumin binders or suboptimal linker design can shorten blood half-life while still maintaining enhanced tumor accumulation relative to non-binder analogs. Although we lack albumin-binding assay and early biodistribution data, the observed pharmacokinetics align with these reports.

In contrast, for Lu-177 labeling, the reaction is known to benefit from prolonged incubation under elevated temperatures (typically 90–95 °C), where efficient labeling occurs through thermal activation rather than mechanical agitation. Our protocol followed standard practice for [^177^Lu]Lu-DOTA complexation, which requires a 60-minute incubation to ensure complete coordination and high metabolic stability. The AHA (aminohexanoic acid) and AMBA (4-aminomethylbenzoic acid) linkers were introduced based on prior structure–activity relationship studies which demonstrated that the physicochemical properties of the linker—such as hydrophilicity, rigidity, and steric hindrance—can significantly influence non-target tissue uptake, particularly in organs such as the liver, kidneys, and stomach [34]. AHA, a flexible aliphatic spacer, introduces hydrophilicity and distance between the chelator and the pharmacophore. This reduces non-specific binding to lipophilic compartments such as hepatic tissue and minimizes renal reabsorption, which often occurs with more hydrophobic constructs. On the other hand, AMBA contributes aromatic rigidity and steric bulk, which has been reported to result in lower binding to off-target proteins and transporters. Additionally, AMBA-based linkers may reduce enzymatic degradation, enhancing in vivo stability and promoting faster clearance from non-target tissues [35]. Experimental data from other previous CCK2R-targeted ligand studies have shown that replacing linear linkers with AHA or AMBA derivatives results in lower uptake in the stomach and kidneys, while maintaining or improving tumor targeting efficacy. Therefore, these linker modifications were implemented with the goal of improving tumor-to-organ ratios and minimizing potential toxicity from non-specific radiation exposure.

Compared with PP-F11N, both INER-PP-F11N analogs showed enhanced binding and internalization in A431 cells expressing CCK2R. Notably, ^177^Lu-DOTA-INER-PP-F11N-1 resulted in 27% higher internalization activity, suggesting that structural modification improved not only stability but also receptor-mediated uptake. In contrast, no significant uptake was observed in CCK2R-negative cells, supporting specificity. This enhancement exceeds previous optimization attempts using PEG spacers or protease inhibitors, where internalization gains were modest [30,35,36,37]. NanoSPECT/CT imaging confirmed selective accumulation of radiotracers in CCK2R-positive tumors. While all compounds demonstrated renal excretion, INER derivatives significantly reduced kidney uptake compared with PP-F11N, which showed prolonged renal retention. This is critical given the severe nephrotoxicity reported with Y-90-labeled PP-F11N [26]. Biodistribution studies further revealed a favorable tumor-to-kidney ratio for INER-PP-F11N-1, positioning it as the superior candidate among the three. Additionally, liver and spleen uptake were minimal, indicating that the improved pharmacokinetic profile did not come at the cost of increased non-specific uptake.

The therapeutic study in xenograft mice demonstrated a clear benefit of the INER compounds. Mice treated with ^177^Lu-DOTA-INER-PP-F11N-1 exhibited the most pronounced tumor growth inhibition (841.33 ± 165.70 mm^3^ at D23), significantly better than PP-F11N (1884.72 ± 333.13 mm^3^). The enhanced therapeutic index can be attributed to higher tumor uptake and lower off-target retention, particularly in the kidneys. This finding supports previous modeling predictions that increasing peptide stability and altering blood kinetics can synergistically enhance tumor response while reducing toxicity [30,38,39,40,41,42]. Histological analysis revealed renal tubular necrosis and inflammation in PP-F11N–treated mice, while no such findings were observed with INER-PP-F11N compounds. The reduced nephrotoxicity likely results from both decreased renal accumulation and reduced in vivo degradation, limiting the presence of nephrotoxic radiometabolites. These results confirm the potential of INER modifications to overcome the primary safety limitation that has restricted the clinical translation of CCK2R-targeted radionuclide therapies. Kidney retention is very high for all CCK2R-targeted radiopharmaceuticals. Regarding the elevated kidney retention of ^177^Lu-DOTA-PP-F11N, we hypothesize that the injected mass of peptide and molar activity were optimized for tumor targeting in our model. However, lower specific activity may increase renal uptake due to saturation effects and altered peptide reabsorption dynamics. Although DOTA is well established for ^177^Lu chelation, small differences in chelator–metal complex stability, plasma protein binding, and hydrophilicity can significantly influence renal clearance. Our formulation was not co-injected with lysine or gelofusine, which are known to reduce kidney uptake of radiolabeled peptides [43,44,45]. We are currently conducting follow-up studies using renal protection protocols to further clarify the retention kinetics and confirm whether this is a formulation- or animal-specific phenomenon. In our biodistribution analysis, bone marrow uptake remained below 0.3% ID/g at all time points, suggesting limited marrow exposure under the current dosing regimen. We are also evaluating fractionated dosing and linker modifications (e.g., cleavable linkers) to further optimize pharmacokinetics and reduce hematologic toxicity risk [45,46,47,48].

Over 90% of medullary thyroid carcinomas overexpress CCK2R, and CCK2R is also expressed in subsets of gliomas, endometrial, and pancreatic cancers [8,49,50]. The observed expression correlates with clinical prognosis, as Kaplan–Meier analyses indicated that high CCK2R levels are associated with worse survival in glioma and endometrial cancer (Appendix A). Hence, the clinical implications of a safe and effective CCK2R-targeted agent extend beyond MTC. Due to the endogenous expression of CCK2R, which mainly focuses on the stomach, it represents a key organ at risk in CCK2R-targeted radioligand therapy. In this initial phase of the study, our primary goal was to confirm target-specific uptake and acute safety. While we evaluated biodistribution and radiotracer uptake in the stomach through gamma counting and imaging, histological analysis of the stomach was not performed at this stage, primarily due to the limited scope of our early-phase toxicological assessment. In follow-up studies currently underway, we are expanding histological evaluations to include the stomach, kidneys, pancreas, and bone marrow, which are known or suspected to express CCK2R or accumulate radiotracers non-specifically. These additional histopathological analyses will help us further characterize organ-specific radiation effects, establish dose limits, and optimize radioprotection strategies for clinical translation. While prior attempts using ^99^ᵐTc, ^68^Ga, or ^64^Cu-labeled compounds focused largely on imaging [19,51], our work demonstrates that INER-PP-F11N derivatives hold promise for theranostic applications, capable of both high-contrast tumor imaging and therapeutic efficacy.

The in vivo studies presented here were designed as a pilot investigation to assess preliminary therapeutic efficacy. Each group included 3–5 animals, and tumor volume was measured in a blinded fashion. Although the number of animals in our therapeutic study was limited, we used five mice per group, in alignment with exploratory study standards in early-phase radiotherapeutic evaluation. We acknowledge that this sample size may not be sufficient for drawing definitive statistical conclusions; therefore, the observed therapeutic effects should be interpreted with caution and primarily as preliminary evidence of biological activity. Furthermore, although the measured tumor uptake of the radiolabeled compound was approximately 1.5% ID/g, we note that similar levels of tumor accumulation were reported to induce measurable therapeutic responses in previous peptide receptor radionuclide therapy (PRRT) studies using high-LET radionuclides such as Lu-177 [52]. It is also important to emphasize that the compound used in this study benefits from prolonged tumor retention and slow clearance from non-target tissues, potentially contributing to higher radiation dose deposition over time, despite initial low uptake values. Nevertheless, future studies will include expanded group sizes, biodistribution-based dosimetry, and longitudinal tumor response assessments to more rigorously validate the therapeutic potential of this agent.

Although this study demonstrated enhanced stability, tumor specificity, and therapeutic efficacy of DOTA-INER-PP-F11N derivatives, several limitations should be addressed in future investigations. First, all in vivo experiments were conducted in subcutaneous xenograft models using A431 cells engineered to overexpress CCK2R. While this model allows for controlled evaluation of radiotracer uptake and therapeutic effects, it does not fully recapitulate the tumor microenvironment or metastatic behavior of native CCK2R-expressing cancers such as medullary thyroid carcinoma or pancreatic adenocarcinoma. Second, although the modified analogs demonstrated improved tumor-to-kidney ratios and reduced histopathological renal toxicity, long-term dosimetry and toxicity profiles are necessary before advancing to clinical applications. Finally, although imaging data showed selective tumor localization of radiotracers, semi-quantitative analysis using standardized uptake value (SUV) or kinetic modeling was not performed. Integrating quantitative PET/SPECT metrics could help refine dose planning for therapy.

## 4. Materials and Methods

### 4.1. Reagents and Cell Lines

DOTA-PP-F11N, DOTA-INER-PP-F11N-1, and DOTA-INER-PP-F11N-2 were purchased from Kelowna International Scientific Inc. (Taiwan). The purchased compounds were accompanied by a certificate of analysis (COA), which confirmed chemical identity and purity (>95%). In addition, we validated their suitability via successful radiolabeling and chromatographic behavior analysis, which were consistent with the expected structure. The HPLC profiles and analytical data are included in the Appendix A (Appendix A). The A431-CCK2R (+) and A431-CCK2R (−) cell lines were kindly provided by Dr. Luigi Aloj; the cells were cultured as previously reported [22]. Briefly, A431-CCK2R (+) and A431-CCK2R (−) cells were both maintained in fresh RPMI-1640 supplemented with 10% FCS, penicillin/streptomycin (50 units/mL), sodium pyruvate (1 mM), L-glutamine (2 mM), and nonessential amino acids (2 mM). The cells were cultured at 37 °C under 5% CO_2_ [53]. The medium and reagents for cell culture were purchased from Invitrogen Life Technologies (Carlsbad, CA, USA). The cells were sub-cultured using trypsin (0.25%; Invitrogen, Carlsbad, CA, USA) and then dissociated at approximately 80% confluence. The culture medium was changed every 2–3 days.

### 4.2. Radioisotope Labeling of CCK2R Analogs

Pharmaceutical-grade indium-111 chloride (^111^InCl_3_) was obtained from NARI, Taoyuan, Taiwan (12.5 mCi in 100 μL of 0.01 M HCl). For radiolabeling, 30 μg of targeting peptide (DOTA-PP-F11N, DOTA-INER-PP-F11N-1, or DOTA-INER-PP-F11N-2) was mixed with 6 mCi of ^111^InCl_3_ in 1 M sodium acetate buffer (pH 6.0) to a final volume of 300 μL. The mixture was incubated at 95 °C for 15 min with gentle shaking (500 rpm). No further purification was performed [54,55]. The radiochemical purity of the ^111^In-labeled peptides was assessed using instant thin-layer chromatography (iTLC) using silica gel strips (iTLC-SG, Pall Corporation, New York, NY, USA) with 0.1 M sodium citrate buffer (pH 5.0) as the mobile phase, and by radio-HPLC.

Lutetium-177 (^177^Lu), a theranostic radionuclide emitting both β- and γ-radiation, was purchased from Isotopia Molecular Imaging (Petach Tikva, Israel). The labeling of the same peptide series with ^177^Lu was carried out as previously described [55]. Briefly, 40 μg of peptide and approximately 1 mCi of ^177^LuCl_3_ were mixed in 0.2 M ascorbic acid buffer (pH 4.8) in a final volume of 150 μL per vial, and incubated at 95 °C for 60 min. The labeling efficiencies of [^177^Lu]Lu-DOTA-PP-F11N, [^177^Lu]Lu-DOTA-INER-PP-F11N-1, and [^177^Lu]Lu-DOTA-INER-PP-F11N-2 were evaluated using iTLC-SG with 10% methanol as the mobile phase, and by radio-HPLC.

### 4.3. Metabolic Stability Analysis

To measure the labeling efficiency, ITLC was performed on glass microfiber chromatography paper impregnated with a silica gel (Agilent Technologies, Santa Clara, CA, USA) as described previously [54,55]. Briefly, the ITLC sheets were developed with 10% methanol as solvent (Rf free Lu-177 = 0, Rf Lu-177-labeled compounds = 1) and measured using a radioactive scanner (AR-2000 radio-TLC 464 Imaging Scanner, Bioscan, France). The sample was mixed with acetonitrile at a 1:1 (*v*/*v*) ratio, followed by centrifugation at 16,000× *g* for 5 min. The resulting supernatant was subsequently analyzed using HPLC with UV (220–466 nm) and radio detector with a Zorbax SB-C18 column (Agilent Technologies, Santa Clara, CA, USA). The flow rate was 0.8 mL/min, with the gradient mobile phase going from 90% A 468 buffer (0.1% TFA in water) and 10% B buffer (0.1% TFA in acetonitrile) to 90% B buffer 469 within 20 min.

### 4.4. Analysis of In Vitro Binding and Internalization Assay

In vitro binding and internalization activities were measured using a gamma counter (WIZARD 1480, Perkin-Elmer) [56]. Briefly, A431-CCK2R (+) and A431-CCK2R (−) cells were seeded on 24-well plates (1 × 10^5^ cells/well, Corning, NY, USA) for 12 h. The plated cells were treated with 30 ± 10 GBq/µmol of [^177^Lu]Lu-DOTA-PP-F11N, [^177^Lu]Lu-DOTA-INER-PP-F11N-1, or [^177^Lu]Lu-DOTA-INER-PP-F11N-2 (1 mL/well) and then incubated at 37 °C for either 1 or 4 h. After incubation, the medium was collected, and the cells were then washed with ice-cold buffer. The medium and washed PBS were collected in tube 0. The washed cells were treated with glycine–HCl buffer (0.1 M, pH 3.0) and incubated for 10 min. After incubation, cells were washed again with glycine–HCl buffer. Both glycine-HCl buffer solutions were collected in tube 1. The washed cells were treated with 1 N NaOH (1 mL/well) and incubated for 10 min for cell lysis. After incubation, the cells were washed once with 1 N NaOH. The cell lysate solution was collected in tube 2. All tubes were analyzed using a gamma counter to detect radioactivity. Data were obtained from at least three independent experiments. The respective formulae for in vitro surface binding activity and internalization were as follows:Surface binding = tube 1/(tube 0 + tube 1 + tube 2) × 100%.Internalization = tube 2/(tube 0 + tube 1 + tube 2) × 100%.

### 4.5. Animal Tumor Model

Female ASID mice (five- to six-week-old) purchased from NARlabs (National Applied Research Laboratories, National Laboratory Animal Center, Taipei, Taiwan) were bred in the animal facility of INER (Taoyuan, Taiwan) at 21–23 °C in a light–dark cycle of 12 h. All the protocols for the animal experiments were in accordance with recommendations for the proper use and care of laboratory animals of the Institute of Nuclear Energy Research (approval number: 110004).

To generate A431-CCK2R (+) and A431-CCK2R (−) tumor-bearing mice models, mice (3 per group) were inoculated subcutaneously with 6 × 10^6^ cells, which were injected into the inner right back leg for A431-CCK2R (+) tumors and the left back leg for A431-CCK2R (−) tumors, respectively. These xenografts were monitored until the tumor volume reached approximately 200–500 mm^3^ [57]. The mice were then randomized to subgroups for further experiments. The formula for calculating tumor volume (V) is V = 4/3πr^3^.

### 4.6. NanoSPECT/CT Imaging

The NanoSPECT/CT^®^ plus scanner system (Mediso Medical Imaging Systems; Budapest, Hungary) was utilized to detect in vivo tumors in the in vivo animal models [57]. Both A431-CCK2R (+) and A431-CCK2R (−) cancer cells were inoculated subcutaneously into the inner right leg and left back leg of the mice. Radiopharmaceuticals—[^111^In]In-DOTA-PP-F11N, [^111^In]In-DOTA-INER-PP-F11N-1, and [^111^In]In-DOTA-INER-PP-F11N-2 (1 mCi)—were injected once the mean tumor volume reached approximately 200–500 mm^3^. Inhalant (1–2% isoflurane) was used to anesthetize the mice during imaging acquisition. X-ray CT images and SPECT images were taken 4, 24, and 48 h after tail-vein injection. NanoSPECT imaging was acquired using nine multi-pinhole gamma detectors with high-resolution collimators. The energy window was set at 171 and 245 KeV ± 10%, the image size was set at 256 × 256, and the field of view was set at 60 mm × 100 mm.

For In-111, dual energy windows were centered at 171 keV and 245 keV with a ±10% window width, in accordance with standard nuclear medicine imaging protocols. These two gamma emissions (171 keV and 245 keV) are the principal photons used for In-111 SPECT imaging and allow for optimal image quality and quantification.

For Lu-177, only 208 keV photons were used for imaging, with an energy window set at 208 keV ± 10%. Although Lu-177 also emits lower abundance photons at 113 keV, the 208 keV photon provides better image quality due to reduced scatter and higher energy resolution. Therefore, the gamma camera energy windows were specifically tailored to the photon emission profiles of each radionuclide to ensure optimal detection efficiency and image accuracy.

### 4.7. Biodistribution Study

Mice were sacrificed via CO_2_ asphyxiation 72 h after intravenous injection of 50–100 μCi (50–100 μL) of radiolabeled DOTA-PP-F11N, DOTA-INER-PP-F11N-1, or DOTA-INER-PP-F11N-2 [58]. Tumors and organs of interest were collected for further pathological examinations. The PerkinElmer 2480 Automatic Gamma Counter (PerkinElmer, Waltham, MA, USA) was used to measure radioactivity; the data are presented as the percentage injected dose per gram of tissue (%ID/g).

### 4.8. Therapeutic Responses to Lu-177-Labeled CCK2R Analogs in CCK2R-Expressing Cancer-Bearing Mice

We investigated therapeutic responses in mice receiving [^177^Lu]Lu-DOTA-PP-F11N, [^177^Lu]Lu-DOTA-INER-PP-F11N-1, or [^177^Lu]Lu-DOTA-INER-PP-F11N-2. A431-CCK2R (+) and A431-CCK2R (−) cells were implanted subcutaneously into the right and left inner hind legs of NOD/SCID mice. Once the tumors reached a volume of 200–500 mm^3,^ the mice were intravenously injected with either 200 μCi of [^177^Lu]Lu-DOTA-PP-F11N, [^177^Lu]Lu-DOTA-INER-PP-F11N-1, [^177^Lu]Lu-DOTA-INER-PP-F11N-2, or PBS (as a non-radioactive control). The NanoSPECT/CT^®^ plus scanner system (Mediso Medical Imaging Systems; Budapest, Hungary) was utilized to detect and image tumors in the mouse models. Tumor volumes were measured twice a week for 4 weeks.

### 4.9. Histopathological Examinations

To investigate the pathological alterations in tumor-bearing mice receiving radiopharmaceuticals, non-target tissues, including tumors, heart, liver, and kidneys, were collected. The specimens were first fixed using 10% buffered neutral formalin and then embedded in paraffin. Sections of 3–5 μm thickness were stained with H&E and examined using light microscopy (40 × and 20 ×).

### 4.10. Statistical Analysis

SigmaPlot 12.5 (Systat Software, Inc., San Jose, CA, USA) and the Statistical Package of Social Studies software (IBM SPSS Statistics 22) were used for data analyses. The chi-square test was used to analyze categorical variables, and ANOVA or Student’s *t*-test was used to analyze continuous variables. Kaplan–Meier curves and log-rank tests were used for survival analysis. Statistical significance was defined as *p* < 0.05.

## 5. Conclusions

This study demonstrates that DOTA-INER-PP-F11N-1 and DOTA-INER-PP-F11N-2 are promising CCK2R-targeted radiopharmaceuticals with improved in vivo stability, enhanced tumor uptake, and reduced renal toxicity compared with [^177^Lu]Lu-DOTA-PP-F11N. The incorporation of these structural moieties may have contributed to the optimization of the pharmacokinetic profile and therapeutic efficacy, potentially through albumin-binding or other mechanisms [28,59]. However, further studies are needed to confirm their mode of action. These findings support further preclinical development toward clinical translation for CCK2R-expressing tumors.

## Figures and Tables

**Figure 1 ijms-26-06565-f001:**
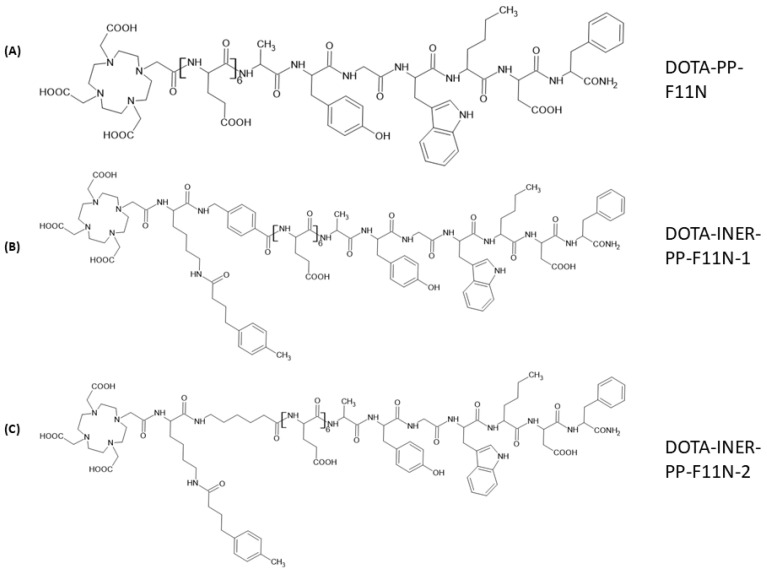
Synthesized CCK2R derivatives with different linkers between DOTA and the C-terminal binding sequence. (**A**) The structure of DOTA-PP-F11N: [DOTA-(Glu)6-Ala-Tyr-Gly-Trp-Nle-Asp-Phe-NH2]; (**B**) the structure of DOTA-INER-PP-F11N-1 [DOTA-Lys(-4-TBA)-6-AMBA-(Glu)6-Ala-Tyr-Gly-Trp-Nle-Asp-Phe-NH2]; and (**C**) the structure of DOTA-INER-PP-F11N-2 [DOTA-Lys(-4-TBA)-AHA-(Glu)6-Ala-Tyr-Gly-Trp-Nle-Asp-Phe-NH2].

**Figure 2 ijms-26-06565-f002:**
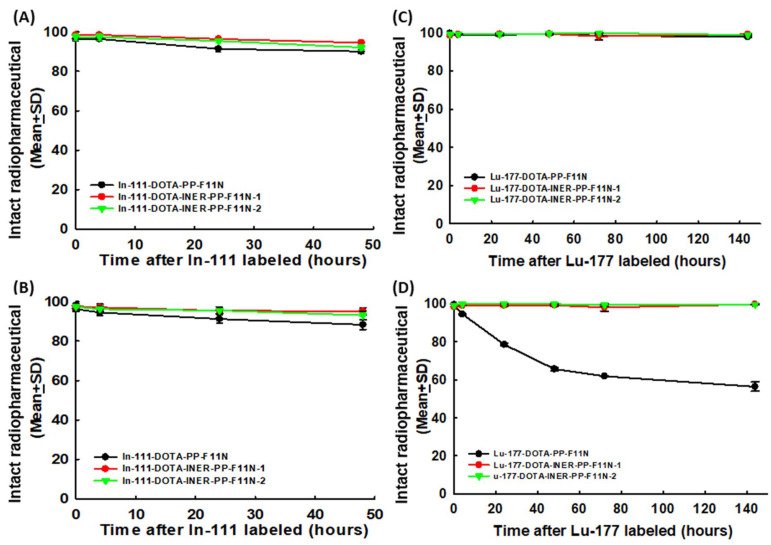
Evaluation of radiopharmaceutical agent using Radio-HPLC. (**A**) Quantitation of metabolic stability of affinity-labeled DOTA-PP-F11N, DOTA-INER-PP-F11N-1, and DOTA-INER-PP-F11N-2 in 4 °C PBS solution at individual time points. (**B**) Quantitation of metabolic stability of In-111-labeled DOTA-PP-F11N, DOTA-INER-PP-F11N-1, and DOTA-INER-PP-F11N-2 in 37 °C FBS-containing solution at individual time points. (**C**) Quantitation of metabolic stability of Lu-177-labeled DOTA-PP-F11N, DOTA-INER-PP-F11N-1, and DOTA-INER-PP-F11N-2 in 4 °C PBS solution at individual time points. (**D**) Quantitation of metabolic stability of Lu-177-labeled DOTA-PP-F11N, DOTA-INER-PP-F11N-1, and DOTA-INER-PP-F11N-2 in 37 °C FBS-containing solution at individual time points. HPLC: high-performance liquid chromatography.

**Figure 3 ijms-26-06565-f003:**
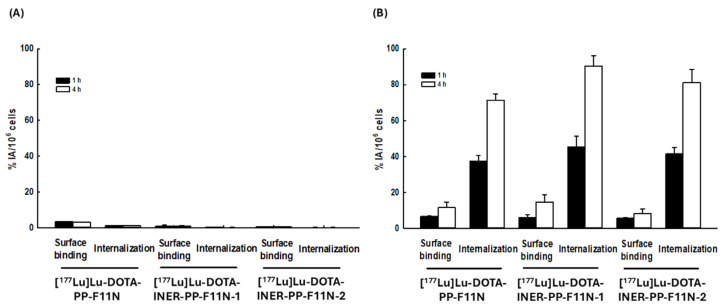
Cellular uptake of [^177^Lu]Lu-DOTA-PP-F11N, [^177^Lu]Lu-DOTA-INER-PP-F11N-1, and [^177^Lu]Lu-DOTA-INER-PP-F11N-2. (**A**) A431-CCK2R (−) cells were treated with Lu-177-labeled CCK2R derivatives and then incubated at 37 °C for 1 or 4 h. Surface binding activity (%IA/106 cells): for [^177^Lu]Lu-DOTA-PP-F11N: 3.42 ± 0.21% (1 h) and 3.24 ± 0.15% (4 h); for [^177^Lu]Lu-DOTA-INER-PP-F11N-1: 1.08 ± 0.49% (1 h) and 0.32 ± 0.10% (4 h); for [^177^Lu]Lu-DOTA-INER-PP-F11N-2: 0.59 ± 0.01% (1 h) and 0.25 ± 0.04% (4 h). Internalization activity (%IA/106 cells): for [^177^Lu]Lu-DOTA-PP-F11N: 1.26 ± 0.06% (1 h) and 1.29 ± 0.01% (4 h); for [^177^Lu]Lu-DOTA-INER-PP-F11N-1: 0.32 ± 0.10% (1 h) and 0.17 ± 0.03 (4 h); for [^177^Lu]Lu-DOTA-INER-PP-F11N-2: 0.25 ± 0.04% (1 h) and 0.15 ± 0.18% (4 h). (**B**) A431-CCK2R (+) cells were treated with Lu-177-labeled CCK2R derivatives and then incubated at 37 °C for 1 or 4 h. Surface binding activity for [^177^Lu]Lu-DOTA-PP-F11N: 6.60 ± 0.17% (1 h) and 11.51 ± 3.16% (4 h); for [^177^Lu]Lu-DOTA-INER-PP-F11N-1: 5.93 ± 1.68% (1 h) and 14.57 ± 4.06% (4 h); for [^177^Lu]Lu-DOTA-INER-PP-F11N-2: 5.51 ± 0.45% (1 h) and 8.11 ± 2.65% (4 h). Internalization activity for [^177^Lu]Lu-DOTA-PP-F11N: 37.37 ± 3.04% (1 h) and 71.44 ± 3.52% (4 h); for [^177^Lu]Lu-DOTA-INER-PP-F11N-1: 45.32 ± 6.12% (1 h) and 90.49 ± 5.65% (4 h); for [^177^Lu]Lu-DOTA-INER-PP-F11N-2: 41.58 ± 3.46% (1 h) and 81.26 ± 7.27% (4 h).

**Figure 4 ijms-26-06565-f004:**
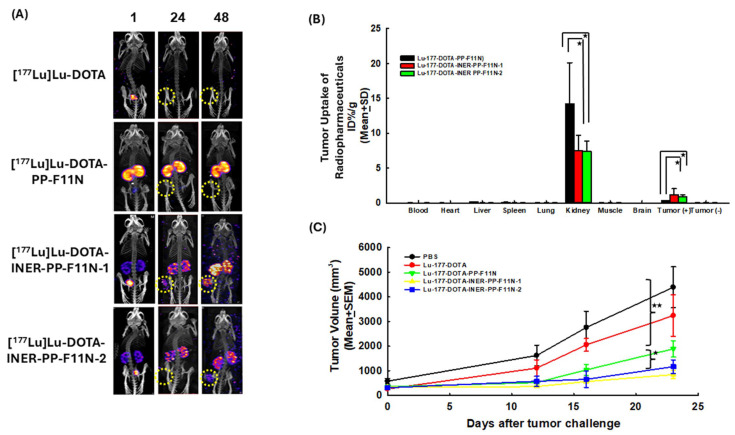
NanoSPECT/CT, biodistribution, and anti-tumor effects of [^177^Lu]Lu-DOTA-CCK2R derivatives in mice. The yellow circle indicated the tumor implant site. (**A**) NanoSPECT/CT images of mice after tail-vein injection of [^177^Lu]Lu-DOTA, [^177^Lu]Lu-DOTA-PP-F11N, [^177^Lu]Lu-DOTA-INER-PP-F11N-1, and [^177^Lu]Lu-DOTA-INER-PP-F11N-2. (**B**) At 72 h after tail-vein injection in tumor-bearing mice, tumor uptake of radiopharmaceuticals (%ID/g) was 0.33 ± 0.02 for [^177^Lu]Lu-DOTA-PP-F11N, 1.14 ± 0.92 for [^177^Lu]Lu-DOTA-INER-PP-F11N-1, and 0.88 ± 0.25 for [^177^Lu]Lu-DOTA-INER-PP-F11N-2. Radiotracer accumulation in the kidneys (%ID/g) was 14.2 ± 5.9 for Lu-177-DTPA-PP-F11N, 7.4 ± 2.3 for [^177^Lu]Lu-DOTA-INER-PP-F11N-1, and 7.4 ± 1.5 for [^177^Lu]Lu-DOTA-INER-PP-F11N-2. (**C**) In vivo tumor growth in mice treated with [^177^Lu]Lu-DOTA-PP-F11N, [^177^Lu]Lu-DOTA-INER-PP-F11N-1, and [^177^Lu]Lu-DOTA-INER-PP-F11N-2. Tumor volume was measured from the day after tumor inoculation. The asterisk (*p* < 0.05) and double asterisks (*p* < 0.01) indicate the significant changes in tumor size.

**Figure 5 ijms-26-06565-f005:**
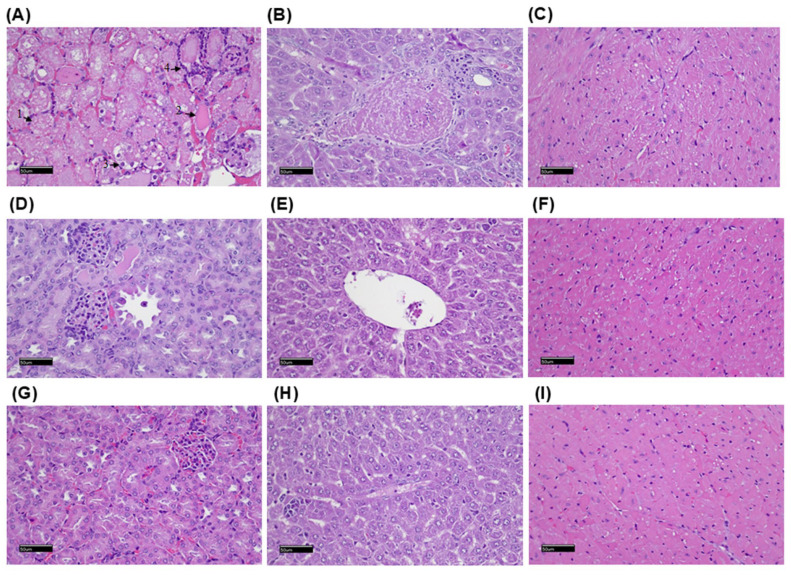
Histopathological examination of the mice. The kidney (**A**), liver (**B**), and heart (**C**) of the mice treated with [^177^Lu]Lu-DOTA-PP-F11N. The kidney showed moderate/severe renal tubular necrosis (arrow 1), moderate renal tubular casts (arrow 2), slight renal tubular degeneration (arrow 3), and slight infiltration of neutrophils (arrow 4). The kidney (**D**), liver (**E**), and heart (**F**) of the mice treated with [^177^Lu]Lu-DOTA-INER-PP-F11N-1. The kidney (**G**), liver (**H**), and heart (**I**) of the mice treated with [^177^Lu]Lu-DOTA-INER-PP-F11N-2 (H&E staining, scale bar = 50 µm).

## Data Availability

The research data in the current study are available from the corresponding author on reasonable request.

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
