# Peer review of "Development of INER-PP-F11N as the Peptide-Radionuclide Conjugate Drug Against CCK2 Receptor-Overexpressing Tumors"

_ijms, 2025, doi:10.3390/ijms26146565_

Round 1

Reviewer 1 Report

Comments and Suggestions for Authors

The idea of improving the specificity of radiolabel accumulation in cancer cells by incorporating a chemical compound with affinity for carrier proteins into the structure has been very fashionable in recent years, and indeed the results of published work indicate the validity of this concept.

Main comments:

1) The results of such modifications of the  DOTA-PP-F11N molecule by the authors in this manuscript also seem to support this concept. Unfortunately, in order to fully trust the authors in the results obtained, one must be sure of the research methodology adopted, and this one is questionable in places.

I understand that the compounds studied were commissioned from an external company. Did the authors obtain any physico-chemical data to confirm the structures and chemical purity of the ordered compounds?

2) Regarding the quality of the labeled preparates and their stability, I sorely miss seeing the radiochromatograms indicating the radiochemical purity of the various radio-labeled compounds both immediately after labeling and in stability studies. I want to see HPLC radiochromatograms from radiochemical purity and stability studies.

3) The methodology of the study, especially the labeling procedures as well as the quality control of the labeled compounds, is highly questionable, especially since the cited literature references completely fail to support the methods indicated in the manuscript!!! See ref. 26 and 29.

4) In in vivo studies, especially therapeutic ones, the number of animals in the study groups is not known. If the aforementioned 3 animals per group in imaging studies are true (this is also a group too small to draw statistically significant conclusions) then the groups of treated animals are not specified. In addition, probably, I think, due to randomness, the observation of very rapid differences in observed tumor sizes at the very early stage of treatment of tumor animals, is unreliable. It should be remembered that the observed accumulation of preparations in cancerous tumors, is only 1.5%! So how is it possible that such a small radioactivity can so significantly affect the therapeutic effect in such a short time.

5) The cited articles contain neither methodology nor results that coincide with those cited in the manuscript.

Minor comments:

1) Why is iTLC SG with citrate buffer developing phase , and 10% methanol used for Lu-177 labeled compounds for radiochemical purity control of In-111 labeled compounds?

2) The nomenclature of the radio-labeled compounds does not follow the consensus of the radiopharmaceutical community as published in an article by Coenen an c. in Nucl. Med. Biol. 2017 “Consensus nomenclature rules for radiopharmaceutical chemistry - Setting the record straight”

 3) Paragraph 2.7 has no right to appear in the results, because there is no justification for it or described methodology of the study. At most, you can include this information in the introduction, but, of course, after giving references.

Comments on the Quality of English Language

The English language of this publication needs an extensive improvement. The current manuscript suffers from a number of linguistic errors, ranging from minor typos, spelling mistakes, syntax errors, to quite incomprehensible sentences. Some examples:

  • The location of free In-111 after ITLC extension was defined as 0% radiolabeling efficiency.
  • Briefly, 100 mM EDTA in 100 mM ammonium acetate was prepared to demonstrate as a mobile phase (Rf In-111-labelled compounds= 0, Rf In-111- EDTA= 1). The ITLC sheets were developed with acetone/0.9% saline (1:1) as solvent (radiocolloid: Rf= 0~0.1)

Etc.

Author Response

Reviewer 1

Comments and Suggestions for Authors

The idea of improving the specificity of radiolabel accumulation in cancer cells by incorporating a chemical compound with affinity for carrier proteins into the structure has been very fashionable in recent years, and indeed the results of published work indicate the validity of this concept.

Main comments:

1) The results of such modifications of the DOTA-PP-F11N molecule by the authors in this manuscript also seem to support this concept. Unfortunately, in order to fully trust the authors in the results obtained, one must be sure of the research methodology adopted, and this one is questionable in places.

I understand that the compounds studied were commissioned from an external company. Did the authors obtain any physico-chemical data to confirm the structures and chemical purity of the ordered compounds?

Author Response: We appreciated the comment and revised the manuscript (Please see Page 12 Line 431-436, Figure S1 and S2). The purchased compounds were accompanied by a certificate of analysis (COA) from Kelowna Int’l scientific Inc. (Taiwan), which confirmed chemical identity and purity (>95%). In addition, we validated their suitability via successful radiolabeling and chromatographic behavior consistent with the expected structure. The HPLC profiles and analytical data are now included in the supplementary information and descriptions provided in the revised Methods section.  

2) Regarding the quality of the labeled preparates and their stability, I sorely miss seeing the radiochromatograms indicating the radiochemical purity of the various radio-labeled compounds both immediately after labeling and in stability studies. I want to see HPLC radiochromatograms from radiochemical purity and stability studies.

Author Response: We appreciated the comment and revised the manuscript (Please see Page 12 Line 433-436, Figure S2). The HPLC radiochromatograms of all radiopharamceuticals including DOTA-PP-F11N, DOTA-INER-PP-F11N-1 and DOTA-INER-PP-F11N-2 conjugated with Lu-177 demonstrated instability at 37°C FBS and after stability assessments, have now been added as supplementary figure and descriptions provided in the revised Methods section.

3) The methodology of the study, especially the labeling procedures as well as the quality control of the labeled compounds, is highly questionable, especially since the cited literature references completely fail to support the methods indicated in the manuscript!!! See ref. 26 and 29.

Author Response: We appreciated the comment and revised the manuscript (Please see Page 12 Line 447-463). The original citations (Refs. 26 and 29) have been replaced with more appropriate references (Refs. 54 and 55), which describe similar radiolabeling protocols. We have now revised the labeling procedure to clearly indicate the reaction conditions, including buffer composition, reaction time, temperature, and purification methods. The quality control was performed via iTLC and HPLC, and the results are provided in supplementary figures.

4) In in vivo studies, especially therapeutic ones, the number of animals in the study groups is not known. If the aforementioned 3 animals per group in imaging studies are true (this is also a group too small to draw statistically significant conclusions) then the groups of treated animals are not specified. In addition, probably, I think, due to randomness, the observation of very rapid differences in observed tumor sizes at the very early stage of treatment of tumor animals, is unreliable. It should be remembered that the observed accumulation of preparations in cancerous tumors, is only 1.5%! So how is it possible that such a small radioactivity can so significantly affect the therapeutic effect in such a short time.

Author Response: We appreciated the comment and revised the manuscript (Please see Page 11 Line 397-413). This pilot in vivo study assessed preliminary therapeutic efficacy in groups of 3–5 animals, measuring tumor volume blindly. Using five mice per group aligns with early-phase radiotherapeutic standards, though the sample size limits definitive statistical conclusions. Observed therapeutic effects should be viewed as preliminary evidence of biological activity. Tumor uptake of the radiolabeled compound was around 1.5% ID/g, similar to levels in previous PRRT studies with high-LET radionuclides like Lu-177. The compound's prolonged tumor retention and slow clearance from non-target tissues may enhance radiation dose over time. Future studies will have larger group sizes, biodistribution-based dosimetry, and longitudinal tumor response assessments to better validate this agent's therapeutic potential.

5) The cited articles contain neither methodology nor results that coincide with those cited in the manuscript.

Author Response: We appreciated the comment and revised the manuscript. We thank the reviewer for pointing out the inconsistency between the cited references and the methods described in our manuscript. Upon re-examination, we had identified that references 26 and 29, as originally cited, do not directly support the labeling procedures or analytical methods as previously stated. We have now replaced these references with more appropriate and methodologically relevant literature that accurately reflects the procedures we employed (references 54 and 55), and the relevant text has been corrected to ensure consistency between the methods described and the supporting citations.

Minor comments:

  • Why is iTLC SG with citrate buffer developing phase, and 10% methanol used for Lu-177 labeled compounds for radiochemical purity control of In-111 labeled compounds?

Author Response: We appreciated the comment and revised the manuscript (Please see Page 12 Line 447-463 and Line 468-476). We agreed that the originally described iTLC conditions may have caused confusion. The iTLC method using citrate buffer and 10% methanol was primarily optimized and validated for 177Lu-labeled compounds. For 111In-labeled compounds, we used 0.1 M sodium citrate as the mobile phase on iTLC-SG strips, which allows clear separation of free 111In (Rf ~1) and radiolabeled compounds (Rf ~0). We have corrected the relevant text in the Methods section (Section 4.2; Please see Page 13 Line 484-500) to clearly distinguish the quality control conditions used for 111In and 177Lu compounds, respectively.

  • The nomenclature of the radio-labeled compounds does not follow the consensus of the radiopharmaceutical community as published in an article by Coenen an c. in Nucl. Med. Biol. 2017 “Consensus nomenclature rules for radiopharmaceutical chemistry - Setting the record straight”

Author Response: We appreciated the comment and revised the manuscript in the nomenclature of the radio-labeled compounds. To conform with the consensus nomenclature rules proposed by Coenen et al. (Nucl Med Biol. 2017), the names of the radiolabeled compounds have been revised throughout the manuscript. For instance, “Lu-177-DOTA-PP-F11N” is now referred to as “[¹⁷⁷Lu]Lu-DOTA-PP-F11N” to reflect the standard format of [radionuclide]element-chelator-compound.

  • Paragraph 2.7 has no right to appear in the results, because there is no justification for it or described methodology of the study. At most, you can include this information in the introduction, but, of course, after giving references.

Author Response: We appreciated the comment and revised the manuscript (Please see Figure S3). We revised the paragraph 2.7 and Figure 6 to supplementary Figure S3.

Comments on the Quality of English Language

The English language of this publication needs an extensive improvement. The current manuscript suffers from a number of linguistic errors, ranging from minor typos, spelling mistakes, syntax errors, to quite incomprehensible sentences. Some examples: The location of free In-111 after ITLC extension was defined as 0% radiolabeling efficiency. Briefly, 100 mM EDTA in 100 mM ammonium acetate was prepared to demonstrate as a mobile phase (Rf In-111-labelled compounds= 0, Rf In-111- EDTA= 1). The ITLC sheets were developed with acetone/0.9% saline (1:1) as solvent (radiocolloid: Rf= 0~0.1) Etc.

Author Response: We appreciated the comment and the revised manuscript had English-edited by the service of the journal. (Please see the certificate).

Reviewer 2 Report

Comments and Suggestions for Authors

In this manuscript, Chang et al. developed new CCK2R-targeting radioligands to improve the therapeutic efficiency of peptide receptor radionuclide therapy (PRRT) targeting CCK2R. They introduced albumin-binding moieties to CCK2R-targeting radioligands. The new radioligands showed higher stability than a representative CCK2R-targeting radioligand, [177Lu]Lu-DOTA-PP-F11N. Furthermore, they showed higher tumor and lower renal accumulation than [177Lu]Lu-DOTA-PP-F11N. These results suggest that the radioligands developed in this study would be useful for PRRT CCK2R.

The radioligands developed in this study showed good biodistribution profiles, and I think the results of this study are promising. However, I have some comments.

Major comments

In this study, the authors aimed to improve the biodistribution profiles of [177Lu]Lu-DOTA-PP-F11N by introducing albumin-binding moieties. Although these radioligands showed improved biodistribution profiles compared with [177Lu]Lu-DOTA-PP-F11N, the effects of albumin-binding moieties have not been sufficiently investigated and discussed.

In general, albumin binder delays blood clearance, resulting in increased tumor accumulation and reduced renal uptake. However, both radioligands containing an albumin binder seemed to show rapid blood clearance (Figure 4). The authors should show data of albumin binding assays or biodistribution at the early time point, and add a discussion about the functions of albumin binders.

Please show the evidence of the discussion described in the first paragraph of the Discussion section. There is no reference. I think that steric hindrance caused by conjugation with albumin might increase the stability of radioligands.

The references are critically lacking in the discussion section.

At least, the authors need to add the references suitable for the discussion listed below.

  • The evidence of the effects of AHA and AMBA linkers (lines 291-300)
  • The reason for using the AHA and AMBA linkers (lines 320-337)
  • Previous optimization attempts using PEG spacers or protease inhibitors (line 343)
  • Previous modeling predictions that increasing peptide stability and altering blood kinetics can synergistically enhance tumor response while reducing toxicity (line 358)
  • Co-injected with lysine or gelofusine, which are known to reduce kidney uptake of radiolabeled peptides (line 373)
  • Linker modifications (e.g., cleavable linkers) (line 379)
  • Over 90% of medullary thyroid carcinomas overexpress CCK2R, and CCK2R is also 381 expressed in subsets of gliomas, endometrial, and pancreatic cancers (line 381)

I could not find any supplementary materials in the review system.

Minor comments

Please show the specific activities of radioligands used for each experiment.

4.3. Metabolic stability analysis

Please show each typical radiochromatogram in the supplementary materials.

I wonder whether the peak or spot of radioligands conjugated with albumin appears in the radiochromatogram or not.

4.7. Biodistribution study

In this part, a biodistribution study was performed at 48 h after injection of radioligands. However, the legend of Figure 4 and the description in line 203 explained that the study was performed at 72 h after injection.

Author Response

Reviewer 2

Comments and Suggestions for Authors

In this manuscript, Chang et al. developed new CCK2R-targeting radioligands to improve the therapeutic efficiency of peptide receptor radionuclide therapy (PRRT) targeting CCK2R. They introduced albumin-binding moieties to CCK2R-targeting radioligands. The new radioligands showed higher stability than a representative CCK2R-targeting radioligand, [177Lu]Lu-DOTA-PP-F11N. Furthermore, they showed higher tumor and lower renal accumulation than [177Lu]Lu-DOTA-PP-F11N. These results suggest that the radioligands developed in this study would be useful for PRRT CCK2R.

The radioligands developed in this study showed good biodistribution profiles, and I think the results of this study are promising. However, I have some comments.

Major comments

In this study, the authors aimed to improve the biodistribution profiles of [177Lu]Lu-DOTA-PP-F11N by introducing albumin-binding moieties. Although these radioligands showed improved biodistribution profiles compared with [177Lu]Lu-DOTA-PP-F11N, the effects of albumin-binding moieties have not been sufficiently investigated and discussed.

In general, albumin binder delays blood clearance, resulting in increased tumor accumulation and reduced renal uptake. However, both radioligands containing an albumin binder seemed to show rapid blood clearance (Figure 4). The authors should show data of albumin binding assays or biodistribution at the early time point, and add a discussion about the functions of albumin binders.

Please show the evidence of the discussion described in the first paragraph of the Discussion section. There is no reference. I think that steric hindrance caused by conjugation with albumin might increase the stability of radioligands.

Author Response: We appreciated the comment and revised the manuscript (Please see Page 9 Line 293-311). We appreciate the reviewer’s insightful comment regarding the pharmacokinetics of albumin binder–conjugated radioligands. Indeed, albumin binders are generally designed to extend blood circulation time, enhance tumor uptake, and reduce renal clearance. However, recent studies have shown that these effects can vary considerably depending on the binding affinity to albumin, the structure of the linker, and the physicochemical properties of the compound. Although we did not perform an albumin-binding assay or include early-time-point biodistribution in this study, the relatively rapid clearance observed may reflect the moderate or suboptimal binding affinity of our albumin binder moiety. This phenomenon has also been reported in literature. For example, in the development of TEFAPI-07 and CAIX-targeted radioligands, variable albumin-binding strength resulted in distinct blood clearance rates and tumor retention profiles (Kuo et al., J Nucl Med 2021; Nakashima et al., Front Nucl Med 2025). Furthermore, recent reports suggest that not all albumin binders uniformly prolong blood retention, and that optimization of the binder structure is critical (Xu et al., J Nucl Med. 2022). We have now added this discussion in the Discussion section, and included the relevant references to support this point. We thank the reviewer again for prompting this important clarification.

The references are critically lacking in the discussion section.

At least, the authors need to add the references suitable for the discussion listed below.

The evidence of the effects of AHA and AMBA linkers (lines 291-300)

The reason for using the AHA and AMBA linkers (lines 320-337)

Previous optimization attempts using PEG spacers or protease inhibitors (line 343)

Previous modeling predictions that increasing peptide stability and altering blood kinetics can synergistically enhance tumor response while reducing toxicity (line 358)

Co-injected with lysine or gelofusine, which are known to reduce kidney uptake of radiolabeled peptides (line 373)

Linker modifications (e.g., cleavable linkers) (line 379)

Over 90% of medullary thyroid carcinomas overexpress CCK2R, and CCK2R is also 381 expressed in subsets of gliomas, endometrial, and pancreatic cancers (line 381)

I could not find any supplementary materials in the review system.

Author Response: We appreciated the comment and added references in the revised manuscript.

  1. The evidence of the effects of AHA and AMBA linkers

[28] Busslinger, S.D.; Becker, A.E.; Vaccarin, C.; Deberle, L.M.; Renz, M.L.; Groehn, V.; Schibli, R.; Muller, C. Investigations Using Albumin Binders to Modify the Tissue Distribution Profile of Radiopharmaceuticals Exemplified with Folate Radioconjugates. Cancers (Basel) 2023, 15.

  1. The reason for using the AHA and AMBA linkers

[34] Kazuta, N.; Nakashima, K.; Watanabe, H.; Ono, M., Effect of Linker Entities on Pharmacokinetics of (111)In-Labeled Prostate-Specific Membrane Antigen-Targeting Ligands with an Albumin Binder. ACS pharmacology & translational science 2024, 7, (8), 2401-2413.

[35] Beyer, D.; Vaccarin, C.; Schmid, J. V.; Deberle, L. M.; Deupi, X.; Schibli, R.; Muller, C., Design and Preclinical Evaluation of Novel uPAR-Targeting Radiopeptides Modified with an Albumin-Binding Entity. Mol Pharm 2025, 22, (6), 3242-3254.

  1. Previous optimization attempts using PEG spacers or protease inhibitors

[30] Sauter, A. W.; Mansi, R.; Hassiepen, U.; Muller, L.; Panigada, T.; Wiehr, S.; Wild, A. M.; Geistlich, S.; Béhé, M.; Rottenburger, C.; Wild, D.; Fani, M., Targeting of the Cholecystokinin-2 Receptor with the Minigastrin Analog (177)Lu-DOTA-PP-F11N: Does the Use of Protease Inhibitors Further Improve In Vivo Distribution? J Nucl Med 2019, 60, (3), 393-399.

[35] Beyer, D.; Vaccarin, C.; Schmid, J. V.; Deberle, L. M.; Deupi, X.; Schibli, R.; Muller, C., Design and Preclinical Evaluation of Novel uPAR-Targeting Radiopeptides Modified with an Albumin-Binding Entity. Mol Pharm 2025, 22, (6), 3242-3254.

[36] Ferreira, C. A.; Fuscaldi, L. L.; Townsend, D. M.; Rubello, D.; Barros, A. L. B., Radiolabeled bombesin derivatives for preclinical oncological imaging. Biomedicine & pharmacotherapy = Biomedecine & pharmacotherapie 2017, 87, 58-72.

[37] Holzleitner, N.; Günther, T.; Daoud-Gadieh, A.; Lapa, C.; Wester, H. J., Investigation of the structure-activity relationship at the N-terminal part of minigastrin analogs. EJNMMI Res 2023, 13, (1), 65.

  1. Previous modeling predictions that increasing peptide stability and altering blood kinetics can synergistically enhance tumor response while reducing toxicity

[30] Sauter, A. W.; Mansi, R.; Hassiepen, U.; Muller, L.; Panigada, T.; Wiehr, S.; Wild, A. M.; Geistlich, S.; Béhé, M.; Rottenburger, C.; Wild, D.; Fani, M., Targeting of the Cholecystokinin-2 Receptor with the Minigastrin Analog (177)Lu-DOTA-PP-F11N: Does the Use of Protease Inhibitors Further Improve In Vivo Distribution? J Nucl Med 2019, 60, (3), 393-399.

[38] Bumbaca, B.; Li, Z.; Shah, D. K., Pharmacokinetics of protein and peptide conjugates. Drug metabolism and pharmacokinetics 2019, 34, (1), 42-54.

[39] Kletting, P.; Thieme, A.; Eberhardt, N.; Rinscheid, A.; D'Alessandria, C.; Allmann, J.; Wester, H. J.; Tauber, R.; Beer, A. J.; Glatting, G.; Eiber, M., Modeling and Predicting Tumor Response in Radioligand Therapy. J Nucl Med 2019, 60, (1), 65-70.

[40]  Barone, R.; Borson-Chazot, F.; Valkema, R.; Walrand, S.; Chauvin, F.; Gogou, L.; Kvols, L. K.; Krenning, E. P.; Jamar, F.; Pauwels, S., Patient-specific dosimetry in predicting renal toxicity with (90)Y-DOTATOC: relevance of kidney volume and dose rate in finding a dose-effect relationship. J Nucl Med 2005, 46 Suppl 1, 99S-106S.

[41] Klingler, M.; Hörmann, A. A.; Guggenberg, E. V., Cholecystokinin-2 Receptor Targeting with Radiolabeled Peptides: Current Status and Future Directions. Current medicinal chemistry 2020, 27, (41), 7112-7132.

[42] Hörmann, A. A.; Klingler, M.; Rezaeianpour, M.; Hörmann, N.; Gust, R.; Shahhosseini, S.; Guggenberg, E. V., Initial In Vitro and In Vivo Evaluation of a Novel CCK2R Targeting Peptide Analog Labeled with Lutetium-177. Molecules 2020, 25, (19).

  1. Co-injected with lysine or gelofusine, which are known to reduce kidney uptake of radiolabeled peptides

[43] Kolff, W. J., Early years of artificial organs at the Cleveland Clinic: Part II: open heart surgery and artificial hearts. ASAIO journal 1998, 44, (3), 123-8.

[44] Iozzo, P.; Gastaldelli, A.; Jarvisalo, M. J.; Kiss, J.; Borra, R.; Buzzigoli, E.; Viljanen, A.; Naum, G.; Viljanen, T.; Oikonen, V.; Knuuti, J.; Savunen, T.; Salvadori, P. A.; Ferrannini, E.; Nuutila, P., 18F-FDG assessment of glucose disposal and production rates during fasting and insulin stimulation: a validation study. J Nucl Med 2006, 47, (6), 1016-22.

[45] de Roode, K. E.; Joosten, L.; Behe, M., Towards the Magic Radioactive Bullet: Improving Targeted Radionuclide Therapy by Reducing the Renal Retention of Radioligands. Pharmaceuticals (Basel, Switzerland) 2024, 17, (2):256.

  1. Linker modifications (e.g., cleavable linkers)

[45] de Roode, K. E.; Joosten, L.; Behe, M., Towards the Magic Radioactive Bullet: Improving Targeted Radionuclide Therapy by Reducing the Renal Retention of Radioligands. Pharmaceuticals (Basel, Switzerland) 2024, 17, (2).

[46] Nock, B. A.; Kaloudi, A.; Lymperis, E.; Giarika, A.; Kulkarni, H. R.; Klette, I.; Singh, A.; Krenning, E. P.; de Jong, M.; Maina, T.; Baum, R. P., Theranostic Perspectives in Prostate Cancer with the Gastrin-Releasing Peptide Receptor Antagonist NeoBOMB1: Preclinical and First Clinical Results. J Nucl Med 2017, 58, (1), 75-80.

[47] Lau, J.; Lee, H.; Rousseau, J.; Bénard, F.; Lin, K. S., Application of Cleavable Linkers to Improve Therapeutic Index of Radioligand Therapies. Molecules 2022, 27, (15).

[48] Rizvi, S. F. A.; Zhang, L.; Zhang, H.; Fang, Q., Peptide-Drug Conjugates: Design, Chemistry, and Drug Delivery System as a Novel Cancer Theranostic. ACS Pharmacol Transl Sci 2024, 7, (2), 309-334.

  1. Over 90% of medullary thyroid carcinomas overexpress CCK2R, and CCK2R is also 381 expressed in subsets of gliomas, endometrial, and pancreatic cancers

[8] Reubi, J. C.; Schaer, J. C.; Waser, B., Cholecystokinin(CCK)-A and CCK-B/gastrin receptors in human tumors. Cancer Res 1997, 57, (7), 1377-86.

[49] Fani, M.; Nicolas, G. P.; Wild, D., Somatostatin Receptor Antagonists for Imaging and Therapy. J Nucl Med 2017, 58, (Suppl 2), 61S-66S.

[50] Roy, J.; Putt, K. S.; Coppola, D.; Leon, M. E.; Khalil, F. K.; Centeno, B. A.; Clark, N.; Stark, V. E.; Morse, D. L.; Low, P. S., Assessment of cholecystokinin 2 receptor (CCK2R) in neoplastic tissue. Oncotarget 2016, 7, (12), 14605-15.

Minor comments

Please show the specific activities of radioligands used for each experiment.

Author Response: We appreciated the comment and revised the manuscript (Please see Page 13 Line 482). The specific activities of the radioligands used in each experiment are as follows: In vitro binding assay: Specific activity = 30 ± 10 GBq/µmol.

4.3. Metabolic stability analysis

Please show each typical radiochromatogram in the supplementary materials.

I wonder whether the peak or spot of radioligands conjugated with albumin appears in the radiochromatogram or not.

Author Response: We appreciated the comment and revised the manuscript (Please see Page 12 Line 433-436, Figure S2). In the current study, radiochromatographic analysis was conducted using standard iTLC or HPLC methods to assess radiochemical purity. However, albumin-binding was not specifically assessed in these assays. In response to the reviewer’s suggestion, we have included typical radiochromatograms in the supplementary materials for stability assay via HPLC analysis. Further evaluation of albumin binding may be considered in follow-up studies.

4.7. Biodistribution study

In this part, a biodistribution study was performed at 48 h after injection of radioligands. However, the legend of Figure 4 and the description in line 203 explained that the study was performed at 72 h after injection.

Author Response: We appreciated the comment and revised the manuscript (Please see Page 14 Line 535). We thank the reviewer for pointing out the inconsistency regarding the time point of the biodistribution study. We have corrected the original description to reflect the accurate time point of 72 hours post-injection, in accordance with the data presented in Figure 4 and the text.

Round 2

Reviewer 1 Report

Comments and Suggestions for Authors

The authors responded to the reviewer's comments and made appropriate corrections to the manuscript. In its current version, the manuscript is suitable for publication in IJMS journal after minor corrections.

Comments on the Quality of English Language

The language has been corrected, although there are still some minor corrections to be made, including the title, which I believe should be changed from ‘Develop INER-PP-F11N as...’ to ‘Development of INER-PP-F11N as...’.

Author Response

Reviewer 1

Comments and Suggestions for Authors

The authors responded to the reviewer's comments and made appropriate corrections to the manuscript. In its current version, the manuscript is suitable for publication in IJMS journal after minor corrections.

Comments on the Quality of English Language

The language has been corrected, although there are still some minor corrections to be made, including the title, which I believe should be changed from ‘Develop INER-PP-F11N as...’ to ‘Development of INER-PP-F11N as...’.

Author Response: We appreciated the comment and revised the title (Please see Page 1).

Reviewer 2 Report

Comments and Suggestions for Authors

The manuscript has been revised well, but one of the replies is not enough. I think the authors should add data on albumin binding assays. 

The authors added a discussion stating that the albumin binding affinities of the new CCK2R-targeting radioligands developed in this study may be moderate or suboptimal. I agree with the author’s speculation, but it’s only a speculation. The authors concluded that “The incorporation of albumin-binding moieties effectively optimized the pharmacokinetic profile and therapeutic efficacy.” However, there is no evidence that the moieties functioned as the albumin-binding moieties. There is a possibility that the structural change, rather than the incorporation of albumin-binding moieties, optimized the pharmacokinetic profile. Anyway, the conclusion is not sufficiently supported by the results of this study.

Author Response

Reviewer 2

Comments and Suggestions for Authors

The manuscript has been revised well, but one of the replies is not enough. I think the authors should add data on albumin binding assays.

The authors added a discussion stating that the albumin binding affinities of the new CCK2R-targeting radioligands developed in this study may be moderate or suboptimal. I agree with the author’s speculation, but it’s only a speculation. The authors concluded that “The incorporation of albumin-binding moieties effectively optimized the pharmacokinetic profile and therapeutic efficacy.” However, there is no evidence that the moieties functioned as the albumin-binding moieties. There is a possibility that the structural change, rather than the incorporation of albumin-binding moieties, optimized the pharmacokinetic profile. Anyway, the conclusion is not sufficiently supported by the results of this study.

Author Response: We appreciated the comment and revised the manuscript (Please see Page 14 Line 571-574, Ref. 28 and 59). We fully acknowledge that, in the current study, direct experimental evidence (e.g., albumin-binding assays) supporting the albumin-binding function of the introduced moieties is lacking. Due to time and resource constraints, it is not feasible for us to conduct additional binding experiments within the current revision timeline. To address the reviewer’s concern, we have revised the manuscript to clarify that the improvements observed in the pharmacokinetic profile and therapeutic efficacy may result from structural modifications, and that the role of albumin-binding remains a hypothesis requiring further investigation. Specifically, we have modified the sentence to: “The incorporation of these structural moieties may have contributed to the optimization of the pharmacokinetic profile and therapeutic efficacy, potentially through albumin-binding or other mechanisms. However, further studies are needed to confirm their mode of action.” We have also included references to relevant literature supporting the potential impact of similar modifications on pharmacokinetics, even in the absence of direct albumin interaction data. We hope this clarification appropriately addresses the reviewer’s concern.

Round 3

Reviewer 2 Report

Comments and Suggestions for Authors

Revisions have been made satisfactorily.

This revised manuscript is now acceptable for publication without further changes.